# DNA microarray analysis of *Staphylococcus aureus* from Nigeria and South Africa

**Adebayo Osagie Shittu**[1,2]*, **Tomiwa Adesoji**[1], **Edet Ekpenyong Udo**[3]

**1** Department of Microbiology, Obafemi Awolowo University, Ile-Ife, Nigeria, **2** Institute of Medical Microbiology, University Hospital Münster, Münster, Germany, **3** Department of Microbiology, Faculty of Medicine, Kuwait University, Safat, Kuwait

* bayo_shittu@yahoo.com

## Abstract

*Staphylococcus aureus* is an important human pathogen with an arsenal of virulence factors and a propensity to acquire antibiotic resistance genes. The understanding of the global epidemiology of *S. aureus* through the use of various typing methods is important in the detection and tracking of novel and epidemic clones in countries and regions. However, detailed information on antibiotic resistance and virulence genes of *S. aureus*, and its population structure is still limited in Africa. In this study, *S. aureus* isolates collected in South Africa (n = 38) and Nigeria (n = 2) from 2001–2004 were characterized by *spa* typing and DNA microarray. The combination of these two methods classified the isolates into seven *spa* types and three clonal complexes (CCs) i.e. t064-CC8 (n = 17), t037-CC8 (n = 8), t1257-CC8 (n = 6), t045-CC5 (n = 5), t951-CC8 (n = 1), t2723-CC88 (n = 1), t6238-CC8 (n = 1), and untypeable-CC8 (n = 1). A high percentage agreement (>95%) and kappa coefficient (>0.60) was largely observed with antibiotic susceptibility testing and DNA microarray, indicating substantial agreement. Some antibiotic and virulence gene markers were associated with specific clones. The detection of the collagen-binding adhesion (*cna*) gene was unique for t037-CC8-MRSA while the enterotoxin gene cluster (*egc*) and staphylococcal complement inhibitor (*scn*) gene were identified with t045-CC5-MRSA. Moreover, the combination of genes encoding enterotoxins (*entA*, *entB*, *entK*, *entQ*) was noted with most of the CC8 isolates. The t045-CC5-MRSA clone was positive for the mercury resistance (*mer*) operon. DNA microarray provides information on antibiotic resistance and virulence gene determinants and can be a useful tool to identify gene markers for specific *S. aureus* clones in Africa.

## Introduction

*Staphylococcus aureus* is a major human pathogen with an array of virulence factors, toxins, and a remarkable ability to acquire antibiotic resistance genes [1, 2]. This capability is further enhanced by the constant emergence of new and diverse clones within regions and countries [3]. The knowledge of the epidemiology of *S. aureus*, particularly of methicillin-resistant *S. aureus* (MRSA), is hinged on the application of various typing methods to assist in tracking newly emerging and epidemic clones [4]. Molecular epidemiological typing tools provide

**Data Availability Statement:** All relevant data are within the manuscript and its Supporting Information files.

**Funding:** AOS: Alexander von Humboldt Foundation ("Georg Forster-

Forschungsstipendium") AOS: Deutsche Forschungsgemeinschaft (SCHA 1994/5-1). AOS: Open Access Publication Fund of the University of Muenster. The funders had no role in study design, data collection and analysis, decision to publish, or preparation of the manuscript.

**Competing interests:** On behalf of other co-authors, I declare no competing interest in the design, implementation, analysis, interpretation of data and write-up of the study titled 'DNA microarray analysis of *Staphylococcus aureus* from Nigeria and South Africa'.

valuable information on the emergence of high-risk pandemic *S. aureus* clones, and the prevalence of antibiotic resistance mechanisms and virulence determinants. This is important in the development of intervention strategies and infection control measures in clinical and non-clinical settings [4].

The *S. aureus* epidemiological landscape in Africa has been described mainly through two molecular typing schemes i.e. *Staphylococcus* protein A (*spa*) typing and multilocus sequence typing (MLST) [5, 6]. These studies revealed that the most widely distributed methicillin-susceptible *S. aureus* (MSSA) clones in Africa include ST5, ST8, ST15, ST30, ST121, and ST152. Whereas ST5, ST30, ST121, and ST152 are predominant in Central and West Africa, ST8, ST15, ST30 are dominant in North Africa [5]. As for MRSA, ST239/241 is a major clone in many African countries, ST8 and ST88 in West, Central and East Africa, ST80 in North Africa, and ST5, ST36 and ST612 in South Africa [5, 6]. However, data on the repertoire of antibiotic resistance and virulence genes of *S. aureus*, and its clonal diversity in Africa are limited. In this study, we characterized archived *S. aureus* isolates from Nigeria and South Africa using DNA microarray. The study aimed to provide detailed information on antibiotic resistance and virulence-related genes, and the population structure of the isolates. This could provide information on antibiotic resistance and virulence genes that may represent epidemiological markers to specific *S. aureus* clones in Africa.

## Materials and methods

### Bacterial isolates

The *S. aureus* isolates have been described in previous investigations [7, 8] and were obtained from different clinical samples from 2001–2004. They comprised mainly archived *S. aureus* from South Africa (MRSA: n = 37; MSSA: n = 1) and two isolates from Nigeria were included based on their phenotypic resistance to cefoxitin and mupirocin, respectively [7]. The isolates (preserved in beads and stored at -80˚C) were sub-cultured on Brain-Heart Infusion Agar (BHIA) plates, re-tested and confirmed (positive coagulase reaction) as *S. aureus* at the MRSA Reference Laboratory, Department of Microbiology, Faculty of Medicine, Kuwait University. Susceptibility of the isolates to penicillin, oxacillin, gentamicin, erythromycin, tetracycline, trimethoprim-sulfamethoxazole, chloramphenicol and mupirocin was performed using the disk diffusion method according to the recommendations of the Clinical Laboratory Standards Institute [9]. *S. aureus* ATCC25923 was utilized as the control strain for antibiotic susceptibility testing (AST).

### DNA isolation

*S. aureus* genomic DNA was obtained from an 18–24 hour old culture on Columbia blood agar. A pre-lysis step as described previously [10] was conducted before proceeding to the protocol of the DNeasy blood and tissue kit (Qiagen Hilden, Germany).

### *Spa* typing and DNA microarray

*Spa* typing was performed by sequencing the hyper-variable region of the protein A gene (*spa*), as described previously [11]. The DNA microarray was performed to screen for the presence of genes for antibiotic resistance, virulence and to assign the isolates to clonal complexes (CCs). Genotyping of the isolates was performed using the *S. aureus* Genotyping Kit 2.0 system (Alere Technologies GmbH, Jena, Germany [now Abbott Rapid Diagnostics GmbH, Jena, Germany]) microarray-based assay. The array covers 334 different targets related to approximately 170 different genes and their allelic variants. The complete list of the target genes,

sequences of probes and primers, and hybridization patterns together with the protocols have been published previously [12–14]. The DNA microarray was performed as described previously [12, 13]. *S. aureus* isolates were cultivated on Colombia blood agar. The DNA extraction was performed using lytic enzymes (lysostaphin, lysozyme, RNase) and buffer from the *S. aureus* Genotyping kit 2.0 and Qiagen DNA extraction kit (Qiagen, Hilden, Germany) according to the manufacturer's instruction. Thereafter, a linear amplification was performed using one primer for each target sequence. During the linear multiplex-amplification, biotin-16-dUTP was incorporated into the amplicons, which were then stringently hybridized to the specific probes on the microarray. After the washing steps, hybridization was detected using streptavidin horseradish peroxidase that triggered local precipitation at those spots where the amplicons were bound. Microarrays were photographed and analysed with a designated reader and software (IconoClust, Alere Technologies). The analysis allowed the detection of certain genes or alleles, as well as assignment to the CCs, and SCC*mec* types. Other target genes include species markers, capsule, agr group typing markers, common antibiotic resistance genes, toxins, microbial surface components recognizing adhesive matrix molecules (MSCRAMMs) and immune evasion cluster. Isolates were assigned to CCs by automated comparison of the microarray hybridization profiles to a large database of previously characterized isolates [13]. The isolates were classified based on the *spa* type and clonal complexes (*spa*-CC).

## Statistical analysis

In the identification of isolates susceptible and resistant to eight antibiotics, the percentage agreement between AST and DNA microarray was calculated from 2 x 2 tables. Furthermore, the level of agreement of the two methods was determined by the Cohen's kappa (κ) test with 95% Confidence Intervals (CI) as described [15] and analyzed using GraphPad Prism (https://www.graphpad.com/quickcalcs/kappa1/). Data was interpreted as follows: no agreement (κ < 0), slight agreement (κ: 0.00–0.20), fair agreement (κ: 0.21–0.40), moderate agreement (κ: 0.41–0.60), substantial agreement (κ: 0.61–0.80), and almost perfect agreement (κ: 0.81–1.00).

## Ethics statement

Ethical clearance was not necessary as archived isolates were analyzed in this study.

## Results

The combination of specific *S. aureus* markers confirmed the identity of the isolates (n = 40) (S1 and S2 Tables). Based on the microarray data, all the isolates harboured genes encoding proteases (*splA*, *splB*, *sspA*, *sspB* and *sspP*), MSCRAMMs (*bbp*, *clfA*, *clfB*, *ebpS*, *fib*, *fnbA*, *map*, *sasG*, *sdrC*, *vwB*), leukocidin (*lukF* and *lukE*), haemolysin (*hlgA*), and intracellular adhesion (*icaA*). However, none possessed the exfoliative toxin (*etA*, *etB*, *etD*), epidermal cell differentiation (*edinA*, *edinB* and *edinC*), surface protein involved in biofilm production (*bap*), and the ACME genes (S4 Table).

AST and DNA microarray were in almost perfect agreement in the detection of isolates susceptible and resistant to chloramphenicol, erythromycin, gentamicin and mupirocin. Moreover, substantial agreement was observed between the two methods in the screening of the isolates against penicillin, oxacillin and tetracycline, while a fair agreement was noted for trimethoprim-sulfamethoxazole (Table 1).

Molecular typing classified the isolates into seven *spa* types t037, t045, t064, t951, t1257, t2723 and t6238, and three clonal complexes (CCs), CC5, CC8 and CC88. The delineation of the various groups (*spa*-CC) and their unique characteristics are described (Fig 1).

**Table 1. Percentage and level of agreement between antibiotic susceptibility testing (AST) and DNA microarray with *S. aureus* isolates from Nigeria and South Africa.**

| Penicillin | | DNA microarray (*blaZ*) | | | Tetracycline | | DNA microarray (*tetK*, *tetM*) | | |
|---|---|---|---|---|---|---|---|---|---|
| | | Negative | Positive | Total | | | Negative | Positive | Total |
| Antibiotic susceptibility testing | Susceptible | 1 | 0 | 1 | Antibiotic susceptibility testing | Susceptible | 4 | 1 | 5 |
| | Resistant | 1 | 38 | 39 | | Resistant | 1 | 34 | 35 |
| | Total | 2 | 38 | 40 | | Total | 5 | 35 | 40 |
| Agreement (%) | | 98% | | | Agreement (%) | | 95% | | |
| Kappa coefficient | | 0.66 | 95% CI: 0.03 to 1.00 | | Kappa coefficient | | 0.77 | 95% CI: 0.47 to 1.00 | |
| Oxacillin | | DNA microarray (*mecA*) | | | Trimethoprim-Sulfamethoxazole | | DNA microarray (*dfrA*) | | |
| | | Negative | Positive | Total | | | Negative | Positive | Total |
| Antibiotic susceptibility testing | Susceptible | 2 | 0 | 2 | Antibiotic susceptibility testing | Susceptible | 6 | 1 | 7 |
| | Resistant | 1 | 37 | 38 | | Resistant | 10 | 23 | 33 |
| | Total | 3 | 37 | 40 | | Total | 16 | 24 | 40 |
| Agreement (%) | | 98% | | | Agreement (%) | | 73% | | |
| Kappa coefficient | | 0.79 | 95% CI: 0.39 to 1.00 | | Kappa coefficient | | 0.37 | 95% CI: 0.10 to 0.64 | |
| Gentamicin | | DNA microarray (*aacA-aphD*, *aphA3*) | | | Chloramphenicol | | DNA microarray (*cat*) | | |
| | | Negative | Positive | Total | | | Negative | Positive | Total |
| Antibiotic susceptibility testing | Susceptible | 3 | 1 | 4 | Antibiotic susceptibility testing | Susceptible | 33 | 0 | 33 |
| | Resistant | 0 | 36 | 36 | | Resistant | 0 | 7 | 7 |
| | Total | 3 | 37 | 40 | | Total | 33 | 7 | 40 |
| Agreement (%) | | 98% | | | Agreement (%) | | 100% | | |
| Kappa coefficient | | 0.84 | 95% CI: 0.55 to 1.00 | | Kappa coefficient | | 1.00 | 95% CI: 1.00 to 1.00 | |
| Erythromycin | | DNA microarray (*ermA*, *ermC*) | | | Mupirocin | | DNA microarray (*mupR*) | | |
| | | Negative | Positive | Total | | | Negative | Positive | Total |
| Antibiotic susceptibility testing | Susceptible | 6 | 1 | 7 | Antibiotic susceptibility testing | Susceptible | 39 | 0 | 39 |
| | Resistant | 0 | 33 | 33 | | Resistant | 0 | 1 | 1 |
| | Total | 6 | 34 | 40 | | Total | 39 | 1 | 40 |
| Agreement (%) | | 98% | | | Agreement (%) | | 100% | | |
| Kappa coefficient | | 0.91 | 95% CI: 0.73 to 1.00 | | Kappa coefficient | | 1.00 | 95% CI: 1.00 to 1.00 | |

Corresponding antibiotic resistance genes detected by DNA microarray are indicated in parenthesis.

## CC5

**t045-CC5 (South German EMRSA or the South German EMRSA/Italian Clone).** Five MRSA isolates belonged to t045. They were grouped with *agr* group II and capsule type 5. While most of them (4/5) possessed the SCC*mec* II element, the cassette chromosome recombinase genes A/B-2 was not detected in one MRSA isolate and was assigned to SCC*mec* type I (S2 Table). All the t045 isolates harboured the resistance genes for aminoglycosides (*aacA-aphD* and *aphA3*), macrolides (*ermA*), fosfomycin (*fosB*), streptothricine (*sat*), and quaternary ammonium compounds (*qacA*). Besides, they were positive for the mercury resistance operon (*mer*). The hybridization signal for *tetK* and *tetM* was absent with one tetracycline-resistant MRSA (SA9). The unique features of this clone include the detection of the enterotoxin gene cluster (*egc*), the presence of only one of the immune evasion cluster (IEC) genes (*scn*),

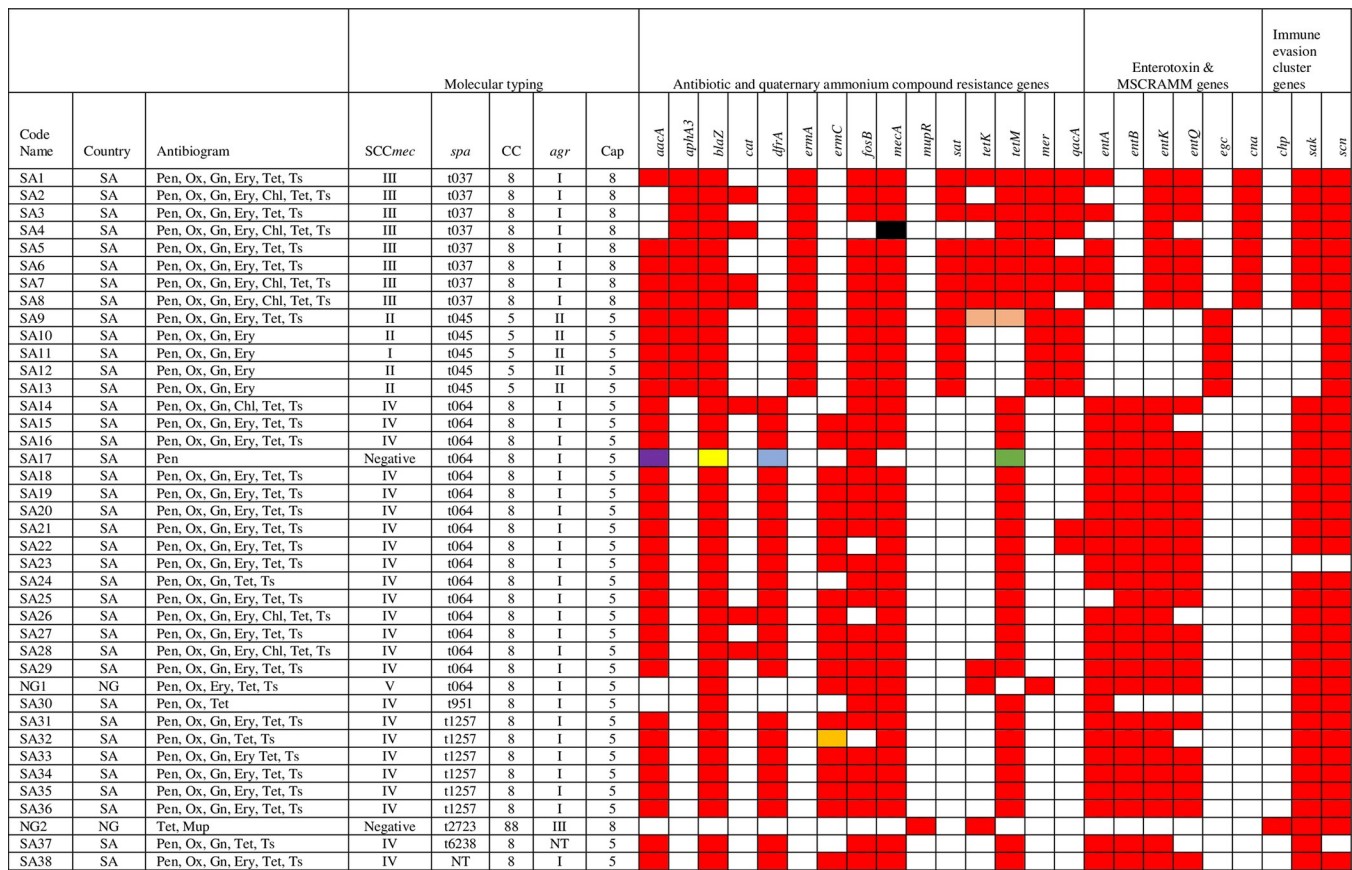

Staphylococcal chromosomal cassette (*mec*); *spa*, *Staphylococcus* protein A; CC, clonal complex; *agr*, accessory gene regulator; cap, capsule; MSCRAMMs, microbial surface components recognizing adhesive matrix molecules; Genes: *aacA*, aminoglycoside acetyltransferase A/aminoglycoside phosphotransferase D; *aphA3*, aminoglycoside phosphotransferase type III; *blaZ*, beta-lactamase; *cat*, chloramphenicol acetyltransferase; *dfrA*, dihydrofolate reductase; *ermA*, *ermC*, rRNA adenine N-6-methyl-transferase; *fosB*, metallothiol transferase; *mecA*, alternate penicillin binding protein 2A; *mupR*, mupirocin resistance; *sat*, streptothricine-acetyltransferase; *tetK*, *tetM* tetracycline-resistance: efflux pump, ribosomal protection, respectively; *mer*, mercury resistance operon; *qac*, quaternary ammonium compounds (drug efflux pump); *entA*, *entB*, *entK*, *entQ* (enterotoxins); *egc*, enterotoxin gene cluster; *cna*, collagen-binding adhesion; *chp*, chemotaxis inhibitory protein; *sak*: staphylokinase; *scn*, staphylococcal complement inhibitor; NT, Non-typeable

□ negative   ■ positive   □ *blaZ* negative; resistant to penicillin   ■ *mecA* negative; resistant to oxacillin

□ *tetK*, *tetM* negative; resistant to tetracycline   □ *ermC* positive; susceptible to erythromycin

□ *tetM* positive; susceptible to tetracycline   □ *aacA-aphD* positive, susceptible to gentamicin

□ *dfrA* positive; susceptible to trimethoprim-sulfamethoxazole

**Fig 1. Antibiotyping and molecular characterization of *S. aureus* isolates from Nigeria and South Africa.**

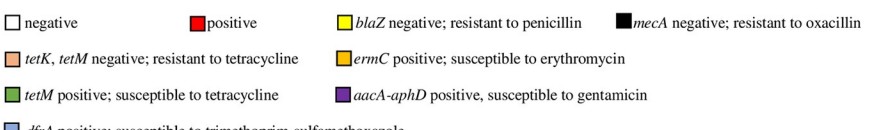

and the lack of the tetracycline resistance (*tetK*, *tetM*) and enterotoxin (*entA*, *entB*, *entK*, *entQ*) genes.

## CC8

The CC8 isolates belonged to five *spa* types, consisting of t064 (n = 17), t037 (n = 8), t1257 (n = 6), t951 (n = 1) and t6238 (n = 1). They were associated with *agr* group I and capsule type 5, except those grouped with *spa* type t037 and assigned to capsule type 8. One MRSA each could not be characterized by *spa* and *agr* typing.

**t037-CC8 (Vienna/Hungarian/Brazilian clone).** The t037 *spa* type was represented by eight isolates. They possessed the SCC*mec* type III genetic element as well as the *mer* (Fig 1)

and the recombinase (*ccrC*) genes (S2 Table). However, one MRSA (SA4) lacked the hybridization signal for *mecA*. All the t037 isolates possessed the *aphA3* and *tetM* genes, and those that exhibited phenotypic resistance to erythromycin (n = 8) and chloramphenicol (n = 4) were positive for the corresponding genes (*ermA* and *cat*) (Fig 1). Seven isolates harboured the fosfomycin and streptothricine resistance determinants (*fosB*, *sat*). The *aacA-aphD* and *tetK* genes were both identified in at least five isolates that were phenotypically resistant to gentamicin and tetracycline, respectively. All the trimethoprim-sulfamethoxazole-resistant MRSA were *dfrA* (dihydrofolate reductase) negative. The enterotoxin genes (*entA*, *entK* and *entQ*) were detected in at least six isolates, while the distinctive feature of this clone was the positive result for the collagen-binding adhesion (*cna*) gene.

**t064-CC8 (USA500).** This clone comprised 16 isolates (MSSA n = 1; MRSA n = 15) from South Africa and MRSA (n = 1) from Nigeria. The SCC*mec* type IV was identified in all the MRSA from South Africa, while the isolate from Nigeria carried the SCC*mec* V element and the *mer* operon (Fig 1). The following genes i.e. *aacA-aphD*, *dfrA*, *ermC*, and *tetM* were detected in at least 14 of the 16 MRSA isolates. Only two MRSA were *qacA*-positive. One MSSA (SA17) that was susceptible (phenotypic) to gentamicin, tetracycline and trimethoprim-sulfamethoxazole was positive for the corresponding resistance genes. The combination of genes encoding enterotoxins (*entA*, *entB*, *entK*, *entQ*) was a common feature noted with most of the isolates.

**t951-CC8 (Lyon Clone/UK-EMRSA-2).** The only MRSA (SCC*mec* IV) possessed the antibiotic resistance genes (*tetM* and *fosB*), and enterotoxin A gene. It was also positive for the immune evasion cluster genes (*sak* and *scn*).

**t1257-CC8 (USA500).** The six isolates belonging to this *spa* type were associated with SCC*mec* type IV. One of the isolates (SA32) was susceptible to erythromycin but yielded a hybridization signal for the *ermC* gene. Moreover, the isolates exhibited similar antibiotic resistance gene profiles (*aacA-aphD*, *dfrA*, *ermC*, and *tetM*), enterotoxin (*entA*, *entB*, *entK*, *entQ*) and immune evasion (*sak*, *scn*) gene content with those assigned with t064-CC8.

**t6238-CC8 (USA500).** The single isolate associated with this *spa* type harboured the SCC*mec* IV element, and the antibiotic resistance (*aacA-aphD*, *dfrA*, *fosB*, *tetM*) and enterotoxin (*entA*, *entB*, *entK*) genes were identified. Furthermore, the isolate was only positive for one of the IEC genes (*sak*). It was negative for *agr* types I-IV.

**spa untypeable-CC8 (USA500).** The gene content of the MRSA isolate was similar to other members of CC8.

**t2723-CC88.** A single MSSA isolate was associated with t2723. It was assigned to *agr* group III and capsule type 8. Phenotypic resistance to tetracycline and mupirocin was confirmed by the detection of the *tetK* and *mupR* genes, respectively. No enterotoxin gene was detected in this isolate. However, it was positive for the IEC (*chp*, *sak*, *scn*) and Panton-Valentine Leukocidin (PVL) genes.

## Discussion

The combination of *spa* typing and DNA microarray was utilized to characterize *S. aureus* isolates obtained in South Africa and Nigeria. The DNA microarray is a DNA-DNA hybridization method containing several probes for the rapid identification, characterization of *S. aureus* resistance and virulence gene profiles, and their assignment into clonal complexes [12]. In the detection of isolates susceptible and resistant to eight antibiotics, substantial to an almost perfect agreement was mainly observed between AST and DNA microarray. The results also revealed the association of some antibiotic resistance gene determinants with certain MRSA clones. Specifically, the *aphA3*, *ermA*, and *mer* genes were unique characteristics

associated with t037-CC8-MRSA and t045-CC5-MRSA (Fig 1). Although t037-CC8-MRSA isolates exhibited resistance to trimethoprim-sulfamethoxazole, they were negative for *dfrA* that encode resistance to trimethoprim in *S. aureus*. While trimethoprim resistance in *S. aureus* can be due to any of three determinants, *dfrA*, *dfrG* and *dfrK*, the *dfrG* is associated with trimethoprim resistance in the majority of the trimethoprim-resistant *S. aureus* in Africa [16, 17]. Trimethoprim-resistant *S. aureus* isolates harbouring *dfrG* and associated with *spa* types t037 and t064 have also been reported in Nigeria [16], which is similar to our findings in this study. This observation suggests that *dfrG* was responsible for trimethoprim resistance in our *dfrA*-negative isolates. The DNA-microarray platform used in this study does not include probes for *dfrG* and explains the reason it was not detected in these isolates.

Interestingly, although t037-CC8-MRSA and t045-CC5-MRSA shared common antibiotic resistance determinants, they differed in the carriage of the tetracycline resistance determinants (*tetK*, *tetM*) that was present in t037-CC8-MRSA and not in t045-CC5-MRSA (Fig 1). MRSA is characterized by the presence of the staphylococcal cassette chromosome *mec* (SCC*mec*), a mobile 21- to 60-kb genetic element, and 13 SCC*mec* types have been identified [18]. The SCC*mec* types II (53.0 kb) and III (66.9 kb) are large elements due to the acquisition and insertion of mobile genetic elements (MBEs). The antibiotic resistance genes observed in the two clones have been identified on MBEs such as transposons including Tn554 (*ermA*), Tn4001 (*aacA-aphD*), Tn5405 (*aphA3*, *sat*), Tn916 (*tetM*), and plasmids i.e. pT181 (*tetK*), pI258 (*mer*) and pNE131 (*ermC*) [19–21]. The t037-CC8-MRSA and t045-CC5-MRSA lineages are typical hospital-associated clones, and their multi-resistant nature are attributed to the various MBEs that harbour different antibiotic resistance genes in the joining regions J1 to J3 [22].

We note with interest that all the t045-CC5-MRSA isolates possessed the mercury resistance operon, a feature also commonly present with t037-CC8-MRSA. The mechanism for the acquisition of SCC*mercury* by *S. aureus* is still unclear although two views have been postulated. The first suggests that this gene determinant may have been integrated into an SCC element with the emergence of SCC*mercury* in coagulase-negative staphylococci, which is subsequently transferred to *S. aureus*. The second opinion is that a plasmid (e.g. pI258) harbouring the resistance gene determinant to the quaternary ammonium compound could have been transferred to *S. aureus* and integrated into an SCC element to form SCC*mercury* [23]. Interestingly, a comparison of our results with a previous report consisting of a collection of CC5-MRSA isolates in the Western Hemisphere [24] revealed that the presence of *mer* gene in t045-CC5-MRSA is a rare feature of this clone. Therefore, future studies are to ascertain whether our observation represents a recent acquisition of the *mer* operon by this lineage. In this study, MRSA with SCC*mec* types IV and V were identified and in addition to β-lactam resistance, the isolates classified as t064/t1257/t6238-CC8 harbouring SCC*mec* IV also possessed genes (*aacA-aphD*, *dfrA*, *ermC* and *tetM*) mediating resistance to aminoglycosides, trimethoprim, macrolides and tetracycline, respectively. The presence of these resistance determinants in our archived isolates support existing data [25, 26] that this multi-resistant lineage is established and well adapted in the hospital environment in South Africa.

Mupirocin is a topical antibiotic that is widely used for nasal decolonization and the prevention of *S. aureus* infections. However, the emergence and increasing rates of resistance and treatment failure are major drawbacks [27]. Two levels of mupirocin resistance have been elucidated i.e. low-level and high-level resistance attributed to various chromosomal mutations, and the acquisition of plasmids (harbouring *mupA* or *mupB* genes), respectively [28, 29]. Decolonization is ineffective with patients and personnel colonized with high-level mupirocin resistant (HmupR) MRSA [27]. Moreover, mupirocin resistance could also facilitate the spread of multidrug resistance through co-selection with other plasmid-borne resistance genes [30,

31]. In this study, the genetic background (t2723-CC88; PVL-positive) of a HmupR MSSA was determined (Fig 1; S2 Table). Only two studies have provided information on the genetic lineage of HmupR *S. aureus* from clinical samples in Africa which include t127, t4805 (MSSA), and t032, t1467 (MRSA) [32, 33]. The prevalence and burden of mupirocin-resistant *S. aureus* are still unclear in many countries in Africa [34]. CC88-MRSA is an established lineage in West, Central and East Africa [5], and the identification of a HmupR-PVL-positive MSSA from this background is worthy of note. Prospective national and continental studies are important to evaluate the prevalence, burden and genetic background of mupR *S. aureus* in Africa.

S. *aureus* produces a range of virulence determinants including at least 23 exotoxins which are categorized into staphylococcal enterotoxins (SEs) comprising SEA-SEE, and staphylococcal enterotoxin-like (SEl) consisting of SEG-SElY [35]. They belong to the family of superantigens (SAgs) with a unique feature to act primarily on the intestine to cause enteritis characterized by emesis [36]. Our investigation indicated that some enterotoxin genes were associated with specific genetic backgrounds, which is in support of previous reports [37, 38]. The t037-CC8-MRSA was characterized by the detection of *entA*, *entK*, and *entQ* genes. The egc cluster (*entG*, *entI*, *entM*, *entN*, *entO*, and *entU*) were associated with t045-CC5-MRSA, while the *entA*, *entB*, *entK* and *entQ* genes were linked with t064/t1257-CC8. The SE genes are carried and disseminated through different MBEs which include prophages, plasmids, transposons, and *S. aureus* pathogenicity islands (SaPIs) [35]. The *entA-entK-entQ* genes are found on the prophage ΦSa3ms and ΦSa3mw, the egc cluster on the genomic island vSaβ and *entB-entK-entQ* have been identified on SaPI3 [39].

## Conclusions

This study characterized archived *S. aureus* isolates from Nigeria and South Africa using two molecular-based typing methods (*spa* typing and DNA microarray). A high level of agreement was observed with AST and DNA microarray. Also, some antibiotic resistance and virulence genes were associated with specific clonal lineages. The *aphA3*, *ermA*, and *mer* genes were associated with hospital-associated clones (t037-CC8-MRSA and t045-CC5-MRSA), *cna* with t037-CC8-MRSA, the *egc* cluster and *scn* with t045-CC5-MRSA, and *entA*, *entB*, *entK*, *entQ* with most of the CC8 isolates. There are some limitations to this study. They include the small and disproportionate number of *S. aureus* analyzed from the two African countries. Moreover, we did not investigate factors that could be responsible for the discrepant results with some isolates based on AST and the microarray assay. The DNA microarray technology has some constraints i.e. high cost of reagents and equipment in resource-limited settings, cross-hybridization reaction, and a moderate level of reproducibility. Nevertheless, the main advantages of the technology include speed compared with procedures involving several PCR and gel electrophoresis, the diverse array of genes investigated, and the quantum of data generated. DNA microarray has provided useful information on gene determinants for antibiotic resistance and virulence, and their relationship with some *S. aureus* genetic background in Africa. Although the outcome of this investigation is not representative of the diverse *S. aureus* clonal lineages in Africa, the genetic markers noted could be a useful adjunct in the molecular typing and tracking of new and emerging *S. aureus* clones on the continent.

## Supporting information

**S1 Table. Raw data on DNA hybridization reaction (indicated as either positive, negative or ambiguous) of the *S. aureus* isolates.**
(XLSX)

**S2 Table. Detailed characteristics of the *S. aureus* isolates (n = 40) including antibiotyping, *spa* types, Clonal complex (CC) affiliation, and DNA microarray hybridization results of some antibiotic resistance and virulence genes.**
(XLS)

**S3 Table. List of target genes and DNA sequences of the probes and primers (including GenBank coordinates) for microarray in *S. aureus* Genotyping Kit 2.0 system.**
(XLS)

**S4 Table. Distribution of selected genes (DNA microarray) in *S. aureus* isolates from Nigeria and South Africa.**
(DOC)

## Acknowledgments

We appreciate the technical assistance of Mrs Tina Verghese, Bindu Mathew and useful comments from Oluyomi Adesoji and Busola Adebusoye.

## Author Contributions

**Conceptualization:** Adebayo Osagie Shittu, Edet Ekpenyong Udo.

**Data curation:** Adebayo Osagie Shittu, Edet Ekpenyong Udo.

**Formal analysis:** Adebayo Osagie Shittu, Edet Ekpenyong Udo.

**Funding acquisition:** Adebayo Osagie Shittu, Edet Ekpenyong Udo.

**Investigation:** Adebayo Osagie Shittu, Edet Ekpenyong Udo.

**Methodology:** Adebayo Osagie Shittu, Edet Ekpenyong Udo.

**Project administration:** Edet Ekpenyong Udo.

**Resources:** Adebayo Osagie Shittu, Edet Ekpenyong Udo.

**Software:** Edet Ekpenyong Udo.

**Supervision:** Edet Ekpenyong Udo.

**Validation:** Adebayo Osagie Shittu, Edet Ekpenyong Udo.

**Visualization:** Adebayo Osagie Shittu, Edet Ekpenyong Udo.

**Writing – original draft:** Adebayo Osagie Shittu, Tomiwa Adesoji, Edet Ekpenyong Udo.

**Writing – review & editing:** Adebayo Osagie Shittu, Tomiwa Adesoji, Edet Ekpenyong Udo.

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
