## [Decision Letter · Decision Letter 0]

26 Oct 2020

PONE-D-20-22325

DNA microarray analysis of Staphylococcus aureus from Nigeria and South Africa

PLOS ONE

Dear Dr. Shittu,

Thank you for submitting your manuscript to PLOS ONE. After careful consideration, we feel that it has merit but does not fully meet PLOS ONE’s publication criteria as it currently stands. Therefore, we invite you to submit a revised version of the manuscript that addresses the points raised during the review process.

Please revise the manuscript taking into account the suggestions and comments of the reivewers. These cahnges will substantially improve the manuscript. Please send the revised version as soon as your convenience.

We look forward to receiving your revised manuscript.

Kind regards,

Monica Cartelle Gestal, PhD

Academic Editor

PLOS ONE

Journal Requirements:

Reviewers' comments:

Reviewer's Responses to Questions

**Comments to the Author**

1. Is the manuscript technically sound, and do the data support the conclusions?

Reviewer #1: Yes

Reviewer #2: No

2. Has the statistical analysis been performed appropriately and rigorously? 

Reviewer #1: I Don't Know

Reviewer #2: No

3. Have the authors made all data underlying the findings in their manuscript fully available?

Reviewer #1: Yes

Reviewer #2: No

4. Is the manuscript presented in an intelligible fashion and written in standard English?

Reviewer #1: Yes

Reviewer #2: Yes

5. Review Comments to the Author

Reviewer #1: This manuscript describes the typing of isolates of Staphylococcus aureus from South Africa and Nigeria. The manuscript is a survey of historic strains from these two countries and compares spa typing to DNA microarray. Overall, the manuscript is well written. As this manuscript is mainly descriptive the analysis is limited but satisfactorily covers the results. More detail of the methods is required to allow for interpretation of the results and conclusions.

Major issues:

1. The methods section lacks the details required to understand what was done in the experiments. How were the strains grown? How was DNA obtained? Did you carry out any statistical analysis? How many biological and technical repeats were carried for each experiment?

2. You have mentioned 2 outliers. With SA32 did you confirm that ermC gene was intact by sequencing? This would tell you if it is a technical problem or not. This approach could also be used for SA17 and blaZ but a negative result would be inconclusive.

Minor:

Line 216: The abbreviation for mupirocin resistance should be MuR throughout.

Line 218: insert full stop after references.

Reviewer #2: Staphylococcus aureus is an important human pathogen with an arsenal of virulence factors and a propensity to acquire antibiotic resistance genes. The understanding of the global epidemiology of S. aureus through the use of various typing methods is important in the detection and tracking of novel and epidemic clones in countries and regions. However, detailed information on antibiotic resistance and virulence genes of S. aureus, and its population structure is still limited in Africa. The current manuscript by Shittu et al. screens S. aureus isolates collected in South Africa (n=38) and Nigeria (n=2) from 2001-2004 for a variety of virulence factors and antimicrobial elements using DNA microarray analysis. This paper basically summarizes the findings of the screen and the concludes that “DNA microarray assay provides information on antibiotic resistance and virulence gene determinants and can be a useful tool to identify gene markers of specific S. aureus clones in Africa”. So while the data reported could be useful, more information and additional analyses are needed for this the report to be of value to the scientific community. Specific comments and suggestions are listed below for consideration:

1) The title is misleading. The current study only includes 40 isolates from Nigeria and only 2 from South Africa. How much value is placed on an “N” of two? To remedy this situation, the number of isolates evaluated for each location should be included in the title. For example “Microarray analysis of 40 Staphylococcus aureus isolates from Nigeria and 2 Staphylococcus aureus isolates South Africa” as one possibility for improvement.

2) Table 1 is confusing. The percentage of selected genes from microarray is not really useful information. Either instead of or in addition to the data provided, the authors should screen the isolates for the presence of a reference nucleotide sequence for each gene. Then report the percent nucleotide identity of each gene in the isolate to the reference nucleotide sequence. This will provide much more useful data!

3) The full nucleotide sequence of each gene in the microarray needs to be provided. Currently no sequence or length of the sequence is provided. Length of the sequence is important because in some instances the length for a microarray target might be only 75 nucleotides. Any search or table for a study of this nature, needs to include the full nucleotide sequence as part of the minimum requirements for publishing!

6. PLOS authors have the option to publish the peer review history of their article (what does this mean?). If published, this will include your full peer review and any attached files.

Reviewer #1: No

Reviewer #2: No

---

## [Author Response · Author response to Decision Letter 0]

31 Dec 2020

Re: Revised manuscript: PONE-D-20-22325 – DNA microarray analysis of Staphylococcus aureus from Nigeria and South Africa

On behalf of other co-authors, I forward herewith the revised manuscript for your kind consideration. It has been formatted to meet the PLOS ONE’s publication criteria. Furthermore, the comments of the reviewers have been examined and we provide a point-by-point response as indicated below.

Journal Requirements:

Response: The revised manuscript has been formatted according to the requirements with PLOS ONE.

Response: Tables 1 and 2 have been included in the revised manuscript. The previous Table 1 is now Supplementary 2.

3. Your ethics statement should only appear in the Methods section of your manuscript. If your ethics statement is written in any section besides the Methods, please move it to the Methods section and delete it from any other section. Please also include your statement in the online submission form via 'Edit submission'.

Response: The ethics statement has been transferred to the Methods section of the manuscript.

Reviewers' comments:

Reviewer #1: This manuscript describes the typing of isolates of Staphylococcus aureus from South Africa and Nigeria. The manuscript is a survey of historic strains from these two countries and compares spa typing to DNA microarray. Overall, the manuscript is well written. As this manuscript is mainly descriptive the analysis is limited but satisfactorily covers the results. More detail of the methods is required to allow for interpretation of the results and conclusions.

Major issues:

1. The methods section lacks the details required to understand what was done in the experiments. How were the strains grown? How was DNA obtained? Did you carry out any statistical analysis? How many biological and technical repeats were carried for each experiment?

Response: We appreciate the comments of the reviewer. Additional information on the strains including DNA isolation is provided in the revised manuscript. The tests were generally not repeated due to their high level of reproducibility. Spa typing and microarray were only repeated in a few instances where the results were ambiguous. We have provided information on some constraints in the conclusion of the manuscript, nevertheless, they do not affect the overall objective of the study.

Statistical analysis: The percentage agreement between the phenotypic (antibiotic susceptibility testing) and genotypic (microarray) methods in the identification of isolates as ‘susceptible’ and ‘resistant’ was calculated from the 2 X 2 tables. Moreover, the Cohen Kappa (κ) test was employed to determine the level of agreement of the two methods. The data is described in the revised manuscript.

2. You have mentioned 2 outliers. With SA32 did you confirm that ermC gene was intact by sequencing? This would tell you if it is a technical problem or not. This approach could also be used for SA17 and blaZ but a negative result would be inconclusive.

Response: We did not conduct a further investigation including sequencing of the blaZ and ermC genes on the isolates that were resistant to penicillin and susceptible to erythromycin, respectively. This is indicated as a limitation of the study in the conclusion. Our focus was to determine the percentage and level of agreement between antibiotic susceptibility testing and DNA microarray. We have provided data (in the revised manuscript) which showed a high level of agreement with the two methods.

Minor:

Line 216: The abbreviation for mupirocin resistance should be MuR throughout.

Response: We are grateful for the comment of the reviewer. However, we prefer the use of mupR (which is widely used) than MuR as suggested.

Line 218: insert full stop after references.

Response: The observation is noted and the amendment is in the revised manuscript.

Reviewer #2: Staphylococcus aureus is an important human pathogen with an arsenal of virulence factors and a propensity to acquire antibiotic resistance genes. The understanding of the global epidemiology of S. aureus through the use of various typing methods is important in the detection and tracking of novel and epidemic clones in countries and regions. However, detailed information on antibiotic resistance and virulence genes of S. aureus, and its population structure is still limited in Africa. The current manuscript by Shittu et al. screens S. aureus isolates collected in South Africa (n=38) and Nigeria (n=2) from 2001-2004 for a variety of virulence factors and antimicrobial elements using DNA microarray analysis. This paper basically summarizes the findings of the screen and the concludes that “DNA microarray assay provides information on antibiotic resistance and virulence gene determinants and can be a useful tool to identify gene markers of specific S. aureus clones in Africa”. So while the data reported could be useful, more information and additional analyses are needed for this the report to be of value to the scientific community. Specific comments and suggestions are listed below for consideration:

1) The title is misleading. The current study only includes 40 isolates from Nigeria and only 2 from South Africa. How much value is placed on an “N” of two? To remedy this situation, the number of isolates evaluated for each location should be included in the title. For example “Microarray analysis of 40 Staphylococcus aureus isolates from Nigeria and 2 Staphylococcus aureus isolates South Africa” as one possibility for improvement.

Response: We appreciate the comments of the reviewer. The small and disproportionate number of S. aureus isolates from the two African countries is a limitation to the study, and this is highlighted in the manuscript. Nevertheless, the origin and number of the isolates are indicated in the abstract, hence we believe it is not necessary to change the title of the manuscript. Moreover, we reiterate that the main objective of the study was to provide detailed information on antibiotic resistance and virulence-related genes, and the population structure of S. aureus isolates (from Africa) based on the microarray assay. We strongly believe that despite the limitation indicated above, the investigation has fulfilled this objective.

2) Table 1 is confusing. The percentage of selected genes from microarray is not really useful information. Either instead of or in addition to the data provided, the authors should screen the isolates for the presence of a reference nucleotide sequence for each gene. Then report the percent nucleotide identity of each gene in the isolate to the reference nucleotide sequence. This will provide much more useful data!

Response: We are grateful for the comment of the reviewer. The microarray assay (Identibac S. aureus genotyping Kit 2.0) is a widely acknowledged tool specifically developed (Alere Technology, Jena, Germany [now Abbott Rapid Diagnostics GmbH, Jena, Germany]) for genotyping of S. aureus isolates. Furthermore, the list of primer/probe sequences has been published previously (Monecke et al., 2011), and this reference is provided in the revised manuscript. 

Monecke S, Coombs G, Shore AC, Coleman DC, Akpaka P, Borg M, Chow H, Ip M, Jatzwauk L, Jonas D, Kadlec K, Kearns A, Laurent F, O'Brien FG, Pearson J, Ruppelt A, Schwarz S, Scicluna E, Slickers P, Tan HL, Weber S, Ehricht R. A field guide to pandemic, epidemic and sporadic clones of methicillin-resistant Staphylococcus aureus. PLoS One. 2011 Apr 6;6(4):e17936. doi: 10.1371/journal.pone.0017936.

Information on the proportion of isolates positive for the antibiotic resistance and virulence genes of S. aureus isolates based on the microarray assay has been described in some recent publications including:

1. Peterson JC, Durkee H, Miller D, Maestre-Mesa J, Arboleda A, Aguilar MC, Relhan N, Flynn HW Jr, Amescua G, Parel JM, Alfonso E. Molecular epidemiology and resistance profiles among healthcare- and community-associated Staphylococcus aureus keratitis isolates. Infect Drug Resist. 2019 Apr 11;12:831-843. doi: 10.2147/IDR.S190245.

2. Senok A, Somily AM, Nassar R, Garaween G, Kim Sing G, Müller E, Reissig A, Gawlik D, Ehricht R, Monecke S. Emergence of novel methicillin-resistant Staphylococcus aureus strains in a tertiary care facility in Riyadh, Saudi Arabia. Infect Drug Resist. 2019 Sep 3;12:2739-2746. doi: 10.2147/IDR.S218870. 

3. Senok A, Nassar R, Celiloglu H, Nabi A, Alfaresi M, Weber S, Rizvi I, Müller E, Reissig A, Gawlik D, Monecke S, Ehricht R. Genotyping of methicillin-resistant Staphylococcus aureus from the United Arab Emirates. Sci Rep. 2020 Oct 29;10(1):18551. doi: 10.1038/s41598-020-75565-w.

4. Udo EE, Boswihi SS, Mathew B, Noronha B, Verghese T, Al-Jemaz A, Al Saqer F. Emergence of Methicillin-Resistant Staphylococcus aureus Belonging to Clonal Complex 15 (CC15-MRSA) in Kuwait Hospitals. Infect Drug Resist. 2020 Feb 21;13:617-626. doi: 10.2147/IDR.S237319. 

5. Boswihi SS, Udo EE, Mathew B, Noronha B, Verghese T, Tappa SB. Livestock-Associated Methicillin-Resistant Staphylococcus aureus in Patients Admitted to Kuwait Hospitals in 2016-2017. Front Microbiol. 2020 Jan 8;10:2912. doi: 10.3389/fmicb.2019.02912.

In our study, data on the proportion of isolates positive or negative for the antibiotic resistance genes (microarray) and antibiotic susceptibility testing (AST) was essential in determining the percentage and level of agreement of the two methods as described in Table 1 (revised manuscript). In conclusion, the previous Table 1 is now presented as Supplementary data (S2 Table).

3) The full nucleotide sequence of each gene in the microarray needs to be provided. Currently no sequence or length of the sequence is provided. Length of the sequence is important because in some instances the length for a microarray target might be only 75 nucleotides. Any search or table for a study of this nature, needs to include the full nucleotide sequence as part of the minimum requirements for publishing!

Response: We appreciate the comment of the reviewer. The list of primer/probe sequences for the microarray has been published previously (Monecke et al., 2011), and this reference is included in the revised manuscript.

Monecke S, Coombs G, Shore AC, Coleman DC, Akpaka P, Borg M, Chow H, Ip M, Jatzwauk L, Jonas D, Kadlec K, Kearns A, Laurent F, O'Brien FG, Pearson J, Ruppelt A, Schwarz S, Scicluna E, Slickers P, Tan HL, Weber S, Ehricht R. A field guide to pandemic, epidemic and sporadic clones of methicillin-resistant Staphylococcus aureus. PLoS One. 2011 Apr 6;6(4):e17936. doi: 10.1371/journal.pone.0017936.

Yours sincerely,

Professor Adebayo Shittu

Corresponding author

---

## [Decision Letter · Decision Letter 1]

27 Jan 2021

PONE-D-20-22325R1

DNA microarray analysis of Staphylococcus aureus from Nigeria and South Africa

PLOS ONE

Dear Dr. Shittu,

Thank you for submitting your manuscript to PLOS ONE. After careful consideration, we feel that it has merit but does not fully meet PLOS ONE’s publication criteria as it currently stands. Therefore, we invite you to submit a revised version of the manuscript that addresses the points raised during the review process.

Reviewer 2 still has some concerns about the methodology and results that are included in this study. Please make sure you carefully revised the manuscript and address all the reviewer comments prior to resubmission.

We look forward to receiving your revised manuscript.

Kind regards,

Monica Cartelle Gestal, PhD

Academic Editor

PLOS ONE

Reviewers' comments:

Reviewer's Responses to Questions

**Comments to the Author**

1. If the authors have adequately addressed your comments raised in a previous round of review and you feel that this manuscript is now acceptable for publication, you may indicate that here to bypass the “Comments to the Author” section, enter your conflict of interest statement in the “Confidential to Editor” section, and submit your "Accept" recommendation.

Reviewer #1: All comments have been addressed

Reviewer #2: (No Response)

2. Is the manuscript technically sound, and do the data support the conclusions?

Reviewer #1: Yes

Reviewer #2: Partly

3. Has the statistical analysis been performed appropriately and rigorously? 

Reviewer #1: Yes

Reviewer #2: Yes

4. Have the authors made all data underlying the findings in their manuscript fully available?

Reviewer #1: Yes

Reviewer #2: No

5. Is the manuscript presented in an intelligible fashion and written in standard English?

Reviewer #1: Yes

Reviewer #2: Yes

6. Review Comments to the Author

Reviewer #1: The authors have addressed all my comments in a satisfactory manor and I have nothing further to suggest.

Reviewer #2: This is the second review for the manuscript by Shittu et al., which describes the screening of S. aureus isolates collected in South Africa (n=38) and Nigeria (n=2) from 2001-2004 for a variety of virulence factors and antimicrobial elements using DNA microarray analysis. This paper basically summarizes the findings of the screen and the concludes that “DNA microarray assay provides information on antibiotic resistance and virulence gene determinants and can be a useful tool to identify gene markers of specific S. aureus clones in Africa”. While the authors have made some revisions to accommodate some of the suggestions posed by reviewers, overall, the authors fail to adequately address some key points raised by the reviewers. The manuscript in its current form is in need of revision to adequately address these key points to improve its suitability for publication.

Major key points:

1) The manuscript in its current form fails to meet PLoS One Data availability requirements. Specifically, “Data: PLOS journals require authors to make all data underlying the findings described in their manuscript fully available without restriction, with rare exception.” Also, the instructions to authors state that all manuscripts incorporating Microarray experiments must meet MIAME (Minimum Information About a Microarray Experiment) requirements. To help aid the authors, MIAME describes the Minimum Information About a Microarray Experiment that is needed to enable the interpretation of the results of the experiment unambiguously and potentially to reproduce the experiment. [Brazma et al. (2001), Nature Genetics].

The six most critical elements contributing towards MIAME are:

a. The raw data for each hybridisation (e.g., CEL or GPR files)

b. The final processed (normalised) data for the set of hybridisations in the experiment (study) (e.g., the gene expression data matrix used to draw the conclusions from the study)

c. The essential sample annotation including experimental factors and their values (e.g., compound and dose in a dose response experiment)

d. The experimental design including sample data relationships (e.g., which raw data file relates to which sample, which hybridisations are technical, which are biological replicates)

e. Sufficient annotation of the array (e.g., gene identifiers, genomic coordinates, probe oligonucleotide sequences or reference commercial array catalog number)

f. The essential laboratory and data processing protocols (e.g., what normalisation method has been used to obtain the final processed data)

The current manuscript fails to provide any of this required information. While references are given, they do not disclose the DNA sequence information of the probes. The manuscript needs to be revised to include all of the above required information! Additionally, providing a supplementary table listing target or gene names, along with the actual DNA sequence of the probes and primers is essential along with MIAME compliance.

2) Lines 81-82: Provide the precise CLSI standard version applied in the phenotpical assay.

3) The authors claim that the objective of the current was to determine the percentage and level of agreement between antibiotic sensitive testing and DNA microarray, which alone is not suitable for publication. The authors also claim that the objective of the current was to provide detailed information on antibiotic resistance and virulence-related genes, and the population structure of S. aureus isolates from Africa. So, which is it? Please clarify! If the second objective is the correct one, then please address the original comments made both reviewers.

7. PLOS authors have the option to publish the peer review history of their article (what does this mean?). If published, this will include your full peer review and any attached files.

Reviewer #1: No

Reviewer #2: No

---

## [Author Response · Author response to Decision Letter 1]

21 Feb 2021

Professor Monica Cartelle Gestal

Academic Editor

PLoS One

Dear Professor Gestal,

Re: Revised manuscript: PONE-D-20-22325-R2 – DNA microarray analysis of Staphylococcus aureus from Nigeria and South Africa

On behalf of other co-authors, I forward herewith the revised manuscript for your kind consideration. The comments of reviewer 2 have been examined and we provide a point-by-point response as indicated below.

Reviewer’s comments

Reviewer #2: This is the second review for the manuscript by Shittu et al., which describes the screening of S. aureus isolates collected in South Africa (n=38) and Nigeria (n=2) from 2001-2004 for a variety of virulence factors and antimicrobial elements using DNA microarray analysis. This paper basically summarizes the findings of the screen and the concludes that “DNA microarray assay provides information on antibiotic resistance and virulence gene determinants and can be a useful tool to identify gene markers of specific S. aureus clones in Africa”. While the authors have made some revisions to accommodate some of the suggestions posed by reviewers, overall, the authors fail to adequately address some key points raised by the reviewers. The manuscript in its current form is in need of revision to adequately address these key points to improve its suitability for publication.

Major key points:

1) The manuscript in its current form fails to meet PLoS One Data availability requirements. Specifically, “Data: PLOS journals require authors to make all data underlying the findings described in their manuscript fully available without restriction, with rare exception.” Also, the instructions to authors state that all manuscripts incorporating Microarray experiments must meet MIAME (Minimum Information About a Microarray Experiment) requirements. To help aid the authors, MIAME describes the Minimum Information About a Microarray Experiment that is needed to enable the interpretation of the results of the experiment unambiguously and potentially to reproduce the experiment. [Brazma et al. (2001), Nature Genetics].

The six most critical elements contributing towards MIAME are:

a. The raw data for each hybridisation (e.g., CEL or GPR files)

b. The final processed (normalised) data for the set of hybridisations in the experiment (study) (e.g., the gene expression data matrix used to draw the conclusions from the study)

c. The essential sample annotation including experimental factors and their values (e.g., compound and dose in a dose response experiment)

d. The experimental design including sample data relationships (e.g., which raw data file relates to which sample, which hybridisations are technical, which are biological replicates)

e. Sufficient annotation of the array (e.g., gene identifiers, genomic coordinates, probe oligonucleotide sequences or reference commercial array catalog number)

f. The essential laboratory and data processing protocols (e.g., what normalisation method has been used to obtain the final processed data)

The current manuscript fails to provide any of this required information. While references are given, they do not disclose the DNA sequence information of the probes. The manuscript needs to be revised to include all of the above required information! Additionally, providing a supplementary table listing target or gene names, along with the actual DNA sequence of the probes and primers is essential along with MIAME compliance.

Response: We appreciate the comments of the reviewer. A description of the microarray assay including the raw data (matrix in excel) has been provided in the revised manuscript. Moreover, in response to the reviewer’s comments, we cite recent publications (kindly see below) that utilized the Staphtype genotyping kit 2.0 and refer to the methodology and data provided in the Supplementary section of these publications. These papers (including those in PLoS One) provided the raw data (in MS-Excel) and clearly showed that the methodology including the DNA sequence for the primers/probes has been published previously. Consequently, we do not agree with the reviewer that a supplementary table with the DNA sequence of the probes and primers should be provided in our manuscript. In conclusion, the description of the genes in S1 Table 1 is stated in S1 Table 3.

Monecke S, Slickers P, Gawlik D, Müller E, Reissig A, Ruppelt-Lorz A, Akpaka PE, Bandt D, Bes M, Boswihi SS, Coleman DC, Coombs GW, Dorneanu OS, Gostev VV, Ip M, Jamil B, Jatzwauk L, Narvaez M, Roberts R, Senok A, Shore AC, Sidorenko SV, Skakni L, Somily AM, Syed MA, Thürmer A, Udo EE, Vremerǎ T, Zurita J, Ehricht R. Molecular Typing of ST239-MRSA-III From Diverse Geographic Locations and the Evolution of the SCCmec III Element During Its Intercontinental Spread. Front Microbiol. 2018 Jul 6;9:1436. doi: 10.3389/fmicb.2018.01436.

Earls MR, Shore AC, Brennan GI, Simbeck A, Schneider-Brachert W, Vremerǎ T, Dorneanu OS, Slickers P, Ehricht R, Monecke S, Coleman DC. A novel multidrug-resistant PVL-negative CC1-MRSA-IV clone emerging in Ireland and Germany likely originated in South-Eastern Europe. Infect Genet Evol. 2019 Apr;69:117-126. doi: 10.1016/j.meegid.2019.01.021. Epub 2019 Jan 21.

Morach M, Käppeli N, Hochreutener M, Johler S, Julmi J, Stephan R, Etter D. Microarray based genetic profiling of Staphylococcus aureus isolated from abattoir byproducts of pork origin. PLoS One. 2019 Sep 6;14(9):e0222036. doi: 10.1371/journal.pone.0222036. 

Alfouzan W, Udo EE, Modhaffer A, Alosaimi A. Molecular Characterization of Methicillin- Resistant Staphylococcus aureus in a Tertiary Care hospital in Kuwait. Sci Rep. 2019 Dec 6;9(1):18527. doi: 10.1038/s41598-019-54794-8.

Boswihi SS, Udo EE, Mathew B, Noronha B, Verghese T, Tappa SB. Livestock-Associated Methicillin-Resistant Staphylococcus aureus in Patients Admitted to Kuwait Hospitals in 2016-2017. Front Microbiol. 2020 Jan 8;10:2912. doi: 10.3389/fmicb.2019.02912. 

Etter D, Corti S, Spirig S, Cernela N, Stephan R, Johler S. Staphylococcus aureus Population Structure and Genomic Profiles in Asymptomatic Carriers in Switzerland. Front Microbiol. 2020 Jun 24;11:1289. doi: 10.3389/fmicb.2020.01289.

Gawlik D, Ruppelt-Lorz A, Müller E, Reißig A, Hotzel H, Braun SD, Söderquist B, Ziegler-Cordts A, Stein C, Pletz MW, Ehricht R, Monecke S. Molecular investigations on a chimeric strain of Staphylococcus aureus sequence type 80. PLoS One. 2020 Oct 14;15(10):e0232071. doi: 10.1371/journal.pone.0232071.

Senok A, Nassar R, Celiloglu H, Nabi A, Alfaresi M, Weber S, Rizvi I, Müller E, Reissig A, Gawlik D, Monecke S, Ehricht R. Genotyping of methicillin resistant Staphylococcus aureus from the United Arab Emirates. Sci Rep. 2020 Oct 29;10(1):18551. doi: 10.1038/s41598-020-75565-w. 

2) Lines 81-82: Provide the precise CLSI standard version applied in the phenotpical assay.

Response: The observation is noted and the amendment is in the revised manuscript.

3) The authors claim that the objective of the current was to determine the percentage and level of agreement between antibiotic sensitive testing and DNA microarray, which alone is not suitable for publication. The authors also claim that the objective of the current was to provide detailed information on antibiotic resistance and virulence-related genes, and the population structure of S. aureus isolates from Africa. So, which is it? Please clarify! If the second objective is the correct one, then please address the original comments made both reviewers.

Response: We appreciate the comment of the reviewer. The statement on the percentage and level of agreement between antibiotic susceptibility testing and DNA microarray has been removed in the revised manuscript (discussion section). The objective of the study is to provide detailed information on antibiotic resistance and virulence-related genes, and the population structure of S. aureus isolates (from Africa) based on the microarray assay.

Yours sincerely,

Professor Adebayo Shittu

Corresponding author

---

## [Decision Letter · Decision Letter 2]

6 Apr 2021

PONE-D-20-22325R2

DNA microarray analysis of Staphylococcus aureus from Nigeria and South Africa

PLOS ONE

Dear Dr. Shittu,

Thank you for submitting your manuscript to PLOS ONE. After careful consideration, we feel that it has merit but does not fully meet PLOS ONE’s publication criteria as it currently stands. Therefore, we invite you to submit a revised version of the manuscript that addresses the points raised during the review process.

There are some minor changes that need to be address, please send us the revised version as early as your convenience

We look forward to receiving your revised manuscript.

Kind regards,

Monica Cartelle Gestal, PhD

Academic Editor

PLOS ONE

Journal Requirements:

Reviewers' comments:

Reviewer's Responses to Questions

**Comments to the Author**

1. If the authors have adequately addressed your comments raised in a previous round of review and you feel that this manuscript is now acceptable for publication, you may indicate that here to bypass the “Comments to the Author” section, enter your conflict of interest statement in the “Confidential to Editor” section, and submit your "Accept" recommendation.

Reviewer #2: (No Response)

2. Is the manuscript technically sound, and do the data support the conclusions?

Reviewer #2: Yes

3. Has the statistical analysis been performed appropriately and rigorously? 

Reviewer #2: Yes

4. Have the authors made all data underlying the findings in their manuscript fully available?

Reviewer #2: No

5. Is the manuscript presented in an intelligible fashion and written in standard English?

Reviewer #2: Yes

6. Review Comments to the Author

Reviewer #2: This is the third review for the manuscript by Shittu et al., which describes the screening of 40 S. aureus isolates collected in South Africa from 2001-2004 for a variety of virulence factors and antimicrobial elements using DNA microarray analysis. This paper basically summarizes the findings of the screen and the concludes that “DNA microarray assay provides information on antibiotic resistance and virulence gene determinants and can be a useful tool to identify gene markers of specific S. aureus clones in Africa”. While the authors have made revisions to accommodate suggestions posed by reviewers, the authors fail to accommodate the request to provide a list of all target or gene names, along with the actual DNA sequence of the probes and primers. Therefore, the manuscript in its current form fails to meet PLoS One Data availability requirements. Instead of simply complying with the policy, the authors choose to argue that 1) previous published papers have not met the policy requirements and 2) that references listed provide the requested information.

This reviewer’s response to these points are:

1) This reviewer was not part of the peer-review process of the noncompliant previous published papers. In contrast, this reviewer is part of the peer-review process for the manuscript by Shittu et al., which in its current form, is NOT compliant.

2) The authors are correct in that one of the cited references provides the required information however, it is provided in a convoluted style. For example, the gene aacA-aphD, a bifunctional enzyme Aac/Aph conferring gentamicin resistance, the probe name is “hp_aacA-59243_PM1” and the probe sequence is “AB049452.1[5151:5178]”.

This reviewer is unsure why the authors persist to refuse to provide a list of all target or gene names, along with the actual DNA sequence of the probes and primers, which would fulfill PLoS One Data availability requirements. These requirements are not meant to be a cumbersome obstacle. They are meant to help make all aspects of the scientific process accessible to everyone.

7. PLOS authors have the option to publish the peer review history of their article (what does this mean?). If published, this will include your full peer review and any attached files.

Reviewer #2: No

---

## [Author Response · Author response to Decision Letter 2]

24 Apr 2021

Re: Revised manuscript: PONE-D-20-22325-R3 – DNA microarray analysis of Staphylococcus aureus from Nigeria and South Africa

On behalf of other co-authors, I forward herewith the revised manuscript for your kind consideration. The Journal requirements and comments of reviewer 2 have been examined and we provide a point-by-point response (in italics) as indicated below.

Journal Requirements

Response: Based on comments/suggestions from the reviewers, the under-listed references were included in the revised manuscript which was not in the original version. 

9. Franklin R. Cockerill, III M, Jean B. Patel, PhD D. M100-S25 Performance Standards for Antimicrobial Susceptibility Testing; Twenty-Fifth Informational Supplement. Clin Lab Stand Inst. 2015.

10. Al-Haqan A, Boswihi SS, Pathan S, Udo EE. Antimicrobial resistance and virulence determinants in coagulase-negative staphylococci isolated mainly from preterm neonates. PLoS One. 2020. doi:10.1371/journal.pone.0236713.

13. Monecke S, Coombs G, Shore AC, Coleman DC, Akpaka P, Borg M, et al. A field guide to pandemic, epidemic and sporadic clones of methicillin-resistant Staphylococcus aureus. PLoS One. 2011. doi:10.1371/journal.pone.0017936.

14. Monecke S, Jatzwauk L, Müller E, Nitschke H, Pfohl K, Slickers P, et al. Diversity of SCCmec elements in staphylococcus aureus as observed in south-eastern Germany. PLoS One. 2016. doi:10.1371/journal.pone.0162654.

15. Landis JR, Koch GG. The Measurement of Observer Agreement for Categorical Data. Biometrics. 1977. doi:10.2307/2529310.

The reference number 10 in the original manuscript: Monecke S, Jatzwauk L, Weber S, Slickers P, Ehricht R. DNA microarray-based genotyping of methicillin-resistant Staphylococcus aureus strains from Eastern Saxony. Clinical Microbiology and Infection. 2008. doi:10.1111/j.1469-0691.2008.01986.x was replaced with the reference below.

12. Monecke S, Slickers P, Ehricht R. Assignment of Staphylococcus aureus isolates to clonal complexes based on microarray analysis and pattern recognition. FEMS Immunol Med Microbiol. 2008. doi:10.1111/j.1574-695X.2008.00426.x.

The reference list is complete and correct in the revised manuscript.

Reviewer’s comments

Reviewer #2: This is the third review for the manuscript by Shittu et al., which describes the screening of 40 S. aureus isolates collected in South Africa from 2001-2004 for a variety of virulence factors and antimicrobial elements using DNA microarray analysis. This paper basically summarizes the findings of the screen and the concludes that “DNA microarray assay provides information on antibiotic resistance and virulence gene determinants and can be a useful tool to identify gene markers of specific S. aureus clones in Africa”. While the authors have made revisions to accommodate suggestions posed by reviewers, the authors fail to accommodate the request to provide a list of all target or gene names, along with the actual DNA sequence of the probes and primers. Therefore, the manuscript in its current form fails to meet PLoS One Data availability requirements. Instead of simply complying with the policy, the authors choose to argue that 1) previous published papers have not met the policy requirements and 2) that references listed provide the requested information.

This reviewer’s response to these points are:

1) This reviewer was not part of the peer-review process of the noncompliant previous published papers. In contrast, this reviewer is part of the peer-review process for the manuscript by Shittu et al., which in its current form, is NOT compliant.

2) The authors are correct in that one of the cited references provides the required information however, it is provided in a convoluted style. For example, the gene aacA-aphD, a bifunctional enzyme Aac/Aph conferring gentamicin resistance, the probe name is “hp_aacA-59243_PM1” and the probe sequence is “AB049452.1[5151:5178]”

This reviewer is unsure why the authors persist to refuse to provide a list of all target or gene names, along with the actual DNA sequence of the probes and primers, which would fulfill PLoS One Data availability requirements. These requirements are not meant to be a cumbersome obstacle. They are meant to help make all aspects of the scientific process accessible to everyone.

Response: We have included the primer and probe names, GenBank coordinates and the corresponding DNA sequences in the revised manuscript (Supplementary data: S1 Table 3). The DNA sequences for the primers and probes were retrieved from the NCBI website https://www.ncbi.nlm.nih.gov/nuccore through their accession numbers and coordinates, as published previously.

Monecke S, Coombs G, Shore AC, Coleman DC, Akpaka P, Borg M, et al. A field guide to pandemic, epidemic and sporadic clones of methicillin-resistant Staphylococcus aureus. PLoS One. 2011. doi: 10.1371/journal.pone.0017936.

Monecke S, Jatzwauk L, Müller E, Nitschke H, Pfohl K, Slickers P, et al. Diversity of SCCmec elements in Staphylococcus aureus as observed in south-eastern Germany. PLoS One. 2016. doi: 10.1371/journal.pone.0162654.

Yours sincerely,

Professor Adebayo Shittu

Corresponding author

---

## [Editor Report · Decision Letter 3]

18 Jun 2021

DNA microarray analysis of Staphylococcus aureus from Nigeria and South Africa

PONE-D-20-22325R3

Dear Dr. Shittu,

We’re pleased to inform you that your manuscript has been judged scientifically suitable for publication and will be formally accepted for publication once it meets all outstanding technical requirements.

Kind regards,

Monica Cartelle Gestal, PhD

Academic Editor

PLOS ONE
---

## [Editor Report · Acceptance letter]

24 Jun 2021

PONE-D-20-22325R3 

DNA microarray analysis of *Staphylococcus aureus* from Nigeria and South Africa 

Dear Dr. Shittu:

I'm pleased to inform you that your manuscript has been deemed suitable for publication in PLOS ONE. Congratulations! Your manuscript is now with our production department. 

Kind regards, 

on behalf of

Dr. Monica Cartelle Gestal 

Academic Editor

PLOS ONE